# Job and life satisfaction among emergency physicians: A qualitative study

**Jesse Kase[1], Benjamin Doolittle**📵[2]*

**1** Yale Program for Medicine, Spirituality and Religion, Yale University School of Medicine, New Haven, Connecticut, United States of America, **2** Internal Medicine & Pediatrics, Yale University School of Medicine, New Haven, Connecticut, United States of America

* benjamin.doolittle@yale.edu

**Data Availability Statement:** All relevant data are within the paper and its Supporting information files.

**Funding:** The authors received no specific funding for this work.

## Abstract

The prevalence of burnout among emergency physicians is among the highest of any specialty. Multiple studies have described factors that contribute to physician burnout, such as age, institutional support, and the electronic medical record (EMR). However, there have been few studies that investigate those physicians who are satisfied with their career and their personal lives. This qualitative study evaluated emergency physicians who were satisfied with both their career and personal lives to propose a model for physician well-being. Physicians were recruited using email solicitation and referral by their peers from June-September 2020. Inclusion criteria involved those physicians who were satisfied with their life and their job and did not meet the criteria for burnout. A qualitative, non-structured interview with open-ended questions was performed with each participant. Emergent themes were identified using standard practice for qualitative studies. Twenty-three physicians participated with a mean age of 45.4 years old (range 32–65), 17 (73.9%) were men, 13 (56.5%) were Caucasian, 6 (26.0%) were Asian/South Asian, 1 (4.3%) were Latino, and 3 (13.0%) were another ethnicity. Several important themes emerged. Physicians satisfied with their lives and their jobs tended to be personally resilient, socially connected, with significant outside interests. These physicians self-identified their personality type as having both introverted and extroverted features. Threats to thriving included ineffective leadership and the EMR. This project proposes a model for job and life satisfaction among emergency physicians. Encouraging these qualities, while fostering supportive leadership, and optimizing the EMR, may improve satisfaction among physicians experiencing burnout.

## Introduction

Burnout is a syndrome characterized by emotional exhaustion, depersonalization, and lack of personal accomplishment [1]. Burnout has been further described as a "syndrome conceptualized as resulting from chronic workplace stress that has not been successfully managed" [2]. Burnout among emergency physicians is often the highest of any specialty, with a prevalence of 60–70% [3].

**Competing interests:** Both authors declare that no competing interests exist.

Much is known about demographic and other factors associated with the burnout syndrome. Emergency physicians suffering from burnout tend to be earlier in their careers, report increased workloads, and have unhealthy work-life balance [4]. Burnout has been associated with increased medical errors, patient and provider dissatisfaction, increased physician turnover, and worse mental health [5, 6]. Costs associated with physician burnout are approximately $7,600 per physician per year and $4.6 billion on a national scale [4]. Several psychosocial factors have been associated with the burnout syndrome among emergency physicians. In a survey of 88 emergency physicians, increased burnout was associated with less freedom and influence at work, less social support, and poorer mental health [7]. Systemic and institutional factors have also been shown to influence the burnout syndrome, such as excessive workload, burdensome work processes, clerical burdens, organizational support, and leadership culture [5].

Given that burnout is well-described in the literature, this project explored a model of physician thriving, as in, those who are not experiencing burnout and are satisfied both in their work and their life. While many studies have examined burnout, few have investigated physician thriving. Thriving can be defined as a physician who is content with their career choice, their family life, and report a minimal amount of burnout days in a year [8]. Physician thriving emphasizes attributes of well-being and happiness and offers a health model rather than a disease model of physician burnout. A physician thriving model can inform individual practice and shape the work environment.

To explore this concept of physician thriving, we employed a qualitative study among emergency physicians who were satisfied with both their lives and their jobs. Qualitative studies are widely recognized as an important method of exploring novel themes [9]. Specifically, this qualitative study explores factors that influence satisfaction and well-being to construct a model for thriving among emergency physicians.

## Methods

### Study design

Yale University Institutional Review Board approved this study. The study number is 2000022828. In addition, permission was granted by the senior leadership of the emergency physician company. In this qualitative study, email and text messages were sent to emergency physicians across the United States who were employed by a privately-held contract management group that staffs several community-based hospitals in the western United States to participate in a study on physician well-being and thriving. Recruitment was also conducted through snowball sampling, where physicians who met inclusion criteria referred other like-minded physicians to the study coordinator [10]. Participation was voluntary and confidential. Both recorded verbal and written consent was obtained at the beginning of the interview. Interested physicians reached out to the study coordinator and were then screened for the burnout syndrome using two validated questions developed by Shanafelt et al. [3]. Physicians who met the criteria for burnout syndrome were not included in the study. Additionally, physicians were screened with two widely-used, validated questions to identify life and career satisfaction: "All things considered, on a scale from 0–10, how satisfied are you with your life?" and "All things considered, on a scale from 0–10, how satisfied are you with your career?" [11]. Those who scored seven or higher for both questions were included in the study, as this correlates with satisfaction levels one standard deviation about the mean or greater [12, 13]. These physicians participated in a semi-structured interview that explored broad themes of personal coping, emotional attitudes, stressors, and social networks.

## Data collection

Physicians were interviewed one-on-one by telephone by the lead author, June-September 2020. Basic demographic information such as age, gender, and ethnicity was collected. The study investigator used a standardized list of open-ended questions to explore themes of job and life satisfaction as an emergency department physician (S1 Appendix). These questions were derived from prior research on thriving and burnout among physicians, and explored dimensions of work environment, social networks, family life, and intrinsic coping such as spirituality and resilience [14–16]. The interviews were recorded and transcribed. Any identifying data was removed from the transcription to preserve anonymity.

## Data analysis

The investigators reviewed the transcripts independently, and identified themes and sub-themes, using a grounded theory-based approach which allowed the capture of naturally occurring themes based on the lived experience of the physicians [9]. Following the initial six interviews, a code structure of emergent themes was developed. With subsequent interviews, we then underwent a collaborative, iterative process of code revision to analyze new themes which were also applied to the original transcripts. Discrepancies were reconciled through a deliberative approach. The themes were then coded to each interview. Thematic saturation was reached after the tenth interview. We continued to interview participants after thematic saturation to explore the possibility of unrevealed topics and to affirm the consistency of identified themes.

## Results

There were 25 interviews conducted with emergency physicians across ten hospitals (Table 1). Two physicians contacted by the interviewer met criteria for burnout and were not included in the study, leaving 23 for data analysis. Eight (35%) of the participants held leadership positions: regional director (1), medical director (5), assistant medical director (2). All participants were board certified in emergency medicine. Of the 23 participants, the mean age was 45.4 years old (range 32–65), with 6.7 years in their present job (range 1–25), 17 (73.9%) were men, 13 (56.5%) were Caucasian, 6 (26.0%) were Asian/South Asian, 1 (4.3%) were Latino, and 3

**Table 1. Characteristics of study participants (n = 23).**

| Characteristic | Value* |
|---|---|
| Mean age (range), years | 45.4 (32–65) |
| Mean time in present job (range) | 6.7 (1–25) |
| Female | 6 (26.01%)) |
| Male | 17 (73.9%) |
| **Ethnic group** | |
| Caucasian | 13 (56.5%) |
| Asian/South Asian | 6 (26.0%) |
| African American | 0 (0.0%) |
| Latino | 1 (4.3%) |
| Other | 3 (13.0%) |
| Satisfaction ("All things considered. . .")** | |
| ". . .how satisfied are you with your life?", mean (SD) | 8.65 (± 0.93) |
| ". . .how satisfied are you with your career?", mean (SD) | 7.46 (± 1.21) |

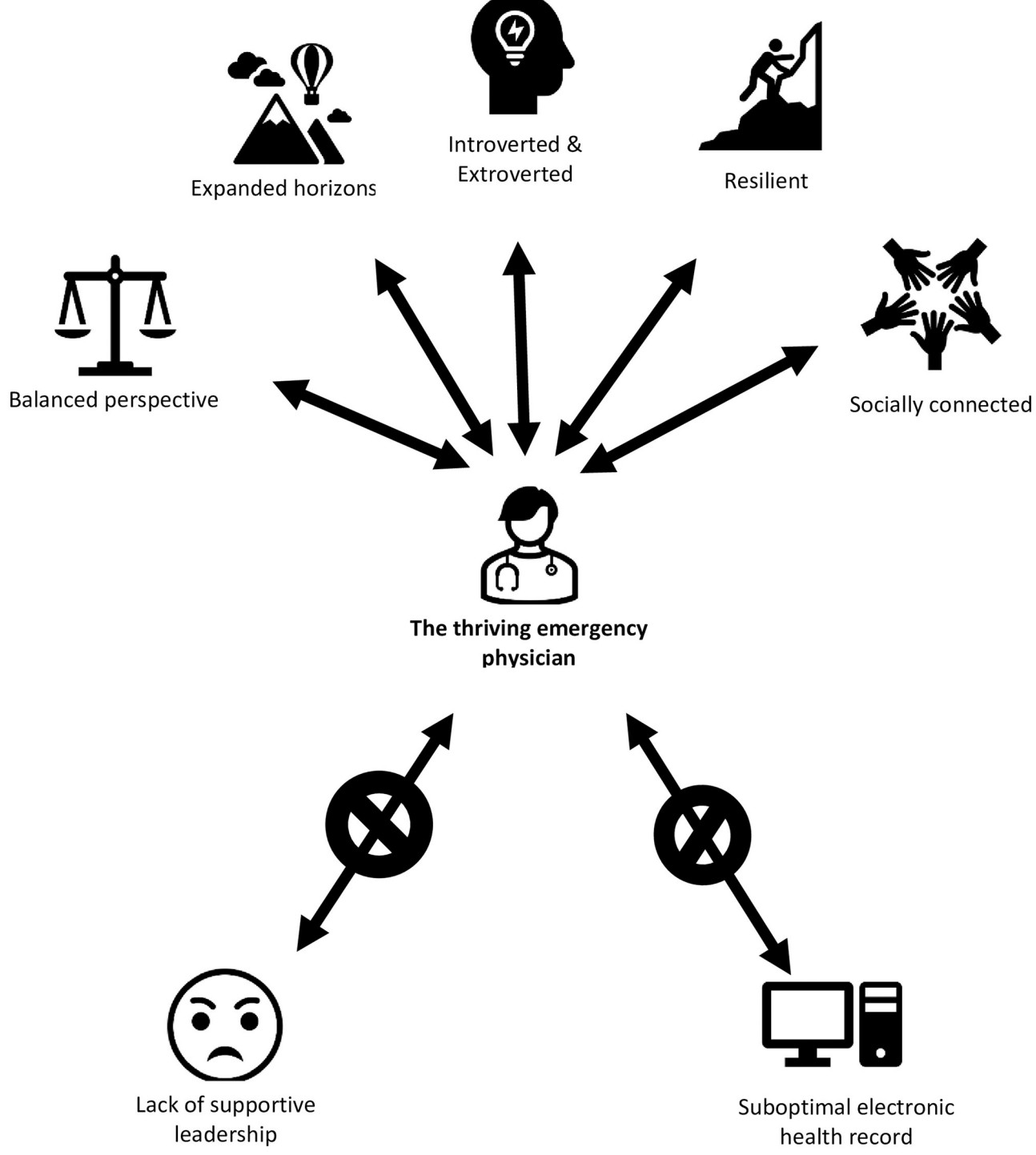

**Fig 1. Conceptual model of emergency physician thriving.**

(13.0%) were another ethnicity (Table 1). Ethnicity and gender were similar to other large studies of emergency physicians [17–19]. Data saturation was reached after the tenth interview. None of the physicians met criteria for burnout. The average score for life satisfaction was 8.65 [sd ± 0.93] and for career satisfaction it was 7.46 [sd ± 2.21].

Five common themes emerged that were associated with thriving as an emergency physician: 1) expanded horizons, 2) social connections, 3) resilience, 4) balance perspective, 5) qualities of both introversion and extroversion. Finally, emergency physicians noted two common threats to their job and life satisfaction: local hospital administration that was not supportive of their role and the EMR (Fig 1).

## Expanded horizons

Emergency physicians across the country expressed how important it is to make sure to have hobbies and activities outside of medicine and work. Most explained one of the main reasons for choosing this field was the ability to pursue interests unrelated to medicine.

"Don't think just about medicine. I would expand your horizons outside of medicine."

"Pick a place where you want to live not the place that pays you the most. Figure out the job and money later. Figure out what you like besides medicine"

"If you going into the ER, you have to pick up some sports. Everyone is out-doorsy so I have become more out-doorsy. I think it is important to have these hobbies because it helps maintain a certain amount of sanity when you are seeing things that are difficult."

"I cook and travel to Europe whenever we can. We live in a place that is very outdoorsy so I do dirt biking and skiing and boating and exercise."

"I just bought a house and enjoying renovating it. I am very big on fitness. Food, beaches, hiking. Very important in my career satisfaction. I picked where I wanted to live before I wanted to work."

Many considered that activities outside of work were a driving factor in their career satisfaction and helped to foster a healthy work-life balance.

## Social connections

Many emergency physicians explained that friendships and other social connections both within and outside the workplace play an important role to career and life satisfaction.

"Connection for me is highest value. Wherever I work they become my family. I like to remember personal things about them."

"Without the human touch, life has no meaning for me. Me being with a friend is soul food."

"Knowing that people are there and they have your back. And it is exciting because you can always talk to someone."

"They are very important. Who is going to help you when you fall?"

"You need to have a good social support."

"Friendships are really really, really important because both helping out friends and being helped out by friends. It re-energizes you and it gives you perspective and has saved my life many times."

### Resilience

When asked about traits that have made these physicians successful at their job, resilience was a common theme.

"You need to have a lot of resilience because you get knocked on your butt a lot doing what we do."

"That is my strongest traits. Once I set a goal, I work towards it until it is reached"

"Having the drive to jump into the fight and finish what you started in addition–perseverance."

Upon being asked what traits are needed to be successful, the physician replied, "Resilience, perseverance, and inner self-confidence." Another replied, "I think being resilient and being able to back bounce on yourself."

### Introverted-extroverted

When asked about personality traits to thriving, physicians described elements of both introversion and extroversion. Many of the emergency physicians called themselves "introverted-extroverts." Participants summarized:

"The introvert side of you gives you your intellectual and moral integrity while the extrovert side of you allows you to be a leader in the department. You need to be both really."

"The introversion helps you to study and do well, and the extroversion helps you be good in a team."

"I think I am fairly balanced. There are periods of my life that I have been more one way than the other. I think in the middle of my life I felt much more extroverted than I was younger, but now during the pandemic, I am quite happy being introverted."

One physician coined the phrase, "ambivert," and shared, "We are in a business of service so you have to be able to communicate and work well with others so you need to be able to do that."

### Balanced perspective

Another theme explored was emotional coping, spirituality and religion. Participants repeatedly invoked the importance of equanimity, maintaining a balanced perspective of the challenges and successes of their role.

"Perspective. Its understanding that things can always be worse and I should be thankful if it is worse for someone else."

"Having an internal perspective makes things a little easier. It just gives you a little different perspective."

"There is something more important than just your career. Get out into the world and you need to have a good work-life balance."

"I am very particular about my schedule. You have to make that balance very deliberate. I have done it well I think."

"Get the work life balance right now. You should work to live instead live to work."

When asked about advice for future doctors, the physician responded, "Don't work yourself to death. Don't live a lifestyle that is like a ball and chain around your neck. Live comfortably within your means."

## Administration as threat

A common threat to thriving expressed by many emergency physicians was a local hospital administration that was unsupportive.

"There are certain things you do as an ER doctor that just degrade you–when it feels like the management above you is just using you to make money."

"I think it has been when I have been working in a situation that is not administered well."

"We are told by lawyers and finance people that we have to violate our core and ethical principles all the time, so being able to navigate that is very important."

"There are certain things you do as an ER doctor that just degrade you–when it feels like the management above you is just using you to make money."

One participant who had previously been in an administrative role expressed regret, "If you are interested in medicine, avoid any administrative roles." When asked when a physician struggled most in his career, he replied, "Administrative things, like if certain processes are slowing me down."

## Electronic medical record and system inefficiencies as a threat

In addition to ineffective administration, another recurrent threat to thriving was the EMR and system inefficiencies. Physicians cited the administrative inefficiencies, charting, and the EMR.

"The EMR is a task I dread every day and probably what I hate the most."

"The EMRs make me struggle." When asked, "When do you struggle the most in your career?" one physician responded, "The EMR. Administrative things like if certain processes are slowing me down." To the same question, another provider responded, "It is more the systems issues that bother me, like the charting, and failure in the system that prevent us from providing care and resources to people." Another physician responded bluntly, "Inefficiency within the ED."

## Discussion

This project explores a model of physician thriving among those working in the Emergency Department of several hospitals among a broad cohort of physicians which generally reflect the ethnic diversity of the national emergency physician workforce [17–19]. Thriving physicians tended to have rich social connections, both within and outside their work environments, as well as notable outside interests and hobbies. Further, thriving physicians had an intrinsically motivated sense of balance, perspective, and resilience. While it is conceptually possible for an emergency physician to experience burnout at work and high satisfaction at home, or vice-versa, this project explored a holistic model of both high job and life satisfaction. The Institute of Healthcare Improvement supplements their Triple Aim of healthcare, quality, efficiency, and patient experience, with a fourth aim, physician work satisfaction [20]. We chose to explore a model of high job and life satisfaction. Our model describes the importance of both work and life satisfaction. Several quantitative projects affirm the importance of a balanced work and home life among physicians [21–23].

There are several implications to this project. These interviews were conducted during a relative trough in the COVID-19 pandemic at many of the hospitals. However, the pandemic still was prevalent during this study. We believe this makes these results even more compelling since these physicians met criteria for thriving and were not burned out during a pandemic. These emergent themes–resilience, social connection, outside interests, with personality types of introversion and extroversion–were associated with career and life satisfaction during a particularly stressful period for the healthcare system. Exploring a model of physician thriving is important, especially since many emergency physicians consider leaving the field, especially women in mid-career [24]. While snowball sampling was the appropriate technique for reaching this specialized subpopulation, limitations should be acknowledged. Namely, these participants were a non-random sample [25]. Physicians may have referred participants in their social network, with similar beliefs and ideas. Thus, there may have been themes that were not captured. However, given the diversity of the participants, with ten hospitals represented, and thematic saturation after ten interviews, we believe our sample population represented valid findings.

Can other physicians learn this model, or is well-being intrinsic to the lives of those interviewed? Some personality traits, such as the balance of introversion and extroversion, may be difficult to teach. However, some personal traits such as gratitude, mindfulness and resilience can be fostered through workshops and other programs. Several intervention studies focused on individual skill-building, such as mindfulness and relaxation techniques, have demonstrated an improvement in well-being and reduced burnout [26–30]. However, these studies tend to have only a few participants and be of short duration [29, 30]. The association was often weak or mixed, highlighting the complex nature of burnout and well-being [29, 30]. Yet, cultivating these intrinsic qualities of resilience, perspective, and expanded horizons may attenuate burnout.

This model affirms the findings by quantitative studies. Shanafelt et al who describes several drivers that leaders need to address to improve physician engagement: resources, workload, control/flexibility, work-life integration, social support, meaning in work, organizational culture [31]. This model affirms the findings derived from the lived experience of emergency physicians in this qualitative study. In particular, optimizing the work environment with appropriate resources and workload as well as emphasizing work-place culture through responsive leadership and social support were key drivers in our model. In addition, Lall et al derived a model of job satisfaction from a survey-based study that identified the importance of reducing burnout, opportunities for non-clinical opportunities, optimizing work-conditions,

family time, and the electronic medical record, and reducing discrimination [19]. Our qualitative study adds validity to the Shanafelt and Lall models [19, 31]. In addition, our study explores intrinsic features such as resilience, balanced perspective, and being both introverted and extroverted.

Interestingly, thriving physicians described their personality type as having both features of introversion and extroversion. The extroversion may help these physicians to lead in moments of high stress, work well in teams, and connect with patients. The introversion may help them to detach, gain perspective, and recharge. This mixed personality trait is complex. Extroversion has been associated with lower emotional exhaustion among physicians [32]. Introversion among nurses has been associated with increased burnout [33]. In a project evaluating leadership efficacy, introverted leaders tend to perform better than extroverts [34]. Either personality type may be positively adaptive or maladaptive to burnout. This mixed personality type–where physicians described features of introversion and extroversion–has not been well-described in the literature and merits further study.

In addition to intrinsic qualities specific to the physician, there are institutional factors that contribute to thriving: the team-oriented spirit of the local environment as well as institutional support from leadership. Support by shift teammates and institutional leaders confer a sense of individual value and personal encouragement crucial for Emergency Department demands. While these physicians met criteria for thriving, some described poor administration and ineffective leadership in their local hospitals as threats to their well-being. In a multispecialty cohort at a large medical center, Shanafelt et al described a significant association between leadership quality and burnout among physicians. Those physicians who scored their immediate supervisor highly had less burnout and increased job satisfaction [35]. In another cohort study, internists with perceived greater institutional support had a higher job and life satisfaction [14]. In another physician cohort, perceived lack of value-alignment by leadership was also associated with increased burnout [36]. Our qualitative study affirmed the need for effective, value-oriented leadership highlighted in these survey projects.

In a systematic review, among eleven studies that associated burnout and leadership styles, "transformational leadership" was associated with lower levels of emotional exhaustion and higher job satisfaction among behavioral health care workers [37]. Transformational leadership is characterized by, "Individualized consideration, intellectual stimulation, inspired motivation, and idealized influence" [37]. In contrast, a "transactional" leadership style has been associated with increased job dissatisfaction and emotional exhaustion among workers. Transactional leaders assign responsibilities to subordinates and tie rewards and punishments based on those responsibilities. While specific data among emergency physicians is lacking, identifying leaders who exhibit a transformational style may enhance the work environment and improve job satisfaction.

In addition, charting in the electronic medical record and other administrative demands were repeatedly noted as negatively impacting physician thriving. This has been repeatedly borne out in a number of studies. Two qualitative studies among primary care physicians affirmed the findings of this project: greater EMR interaction was association with worse job satisfaction [14, 15]. In a multi-clinic cohort study, primary care physicians with more EMR duties had greater burnout and lower job satisfaction than those with fewer EMR duties [38]. Another physician cohort study from all specialties showed that 70% experienced stress related to technology. More time with the EMR was independently associated with stress and burnout [39]. Optimizing the EMR with improved training or more intuitive processes may attenuate job dissatisfaction. This is an important leading-edge issue for physician well-being that merits intervention trials.

In our qualitative study, these physicians did not cite support derived from professional societies, annual meetings, or other continuing education programs (CME). One study did show that participation in CME's was associated with less job stress and positive job satisfaction [40]. Of note, our participants did not cite organized religious involvement and other formal social clubs as important to well-being, which has been noted in broader population-based studies [41]. The social connections among these physicians were mostly along informal networks and friend groups rather than intentional participation in social organizations. Friendships have been associated with greater job satisfaction and reduced burnout among physicians but is not widely studied [14]. Religious involvement and spirituality have been widely studied in the general population and found to be associated with many positive mental health benefits, such as depression, anxiety, and burnout [42, 43]. However, religious involvement and spirituality specifically among physicians has not been well studied. One large cohort project among primary care physicians and psychiatrists demonstrated an association with religious involvement and "a sense of call," but mental health parameters were not investigated [44]. Further study in friendships, religiosity, and other social networks of physicians is warranted.

## Strengths & limitations

Limitations for this project should be considered. First, the physicians in this study were employees of one physician management company which staff many different community-based Emergency Departments in the western United States. While this allowed for heterogeneity of perspective among several local institutions, the national leadership was the same. Thus, these findings may not be generalizable to all emergency physicians, especially those in academic medicine as this was a community-based study. In expressing opinions about leadership, comments reflected the local milieu. Second, while qualitative studies are designed to elicit broad opinions in an open-ended fashion, there is always the possibility of asking leading or biased questions designed to elicit desired responses. Third, *eight of the participants held leadership positions. This may represent a confounding variable. They may experience extra pressure in their leadership roles or less pressure since their administrative duties lessen their direct-facing patient care. However, their responses resonated with non-leaders with the one difference being that the did not regard leadership as a threat. Fourth*, since this study explicitly explored physician well-being, participants may have given answers to satisfy the study purpose. Of note, this project was designed to explore a model of thriving through the lived experience of front-line emergency physicians. It was not designed to explore factors related to burnout. Fifth, while most physicians were male and Caucasian, our sample reflects similar demographics of larger, quantitative studies [10–12]. However, given the diversity of participants, from different local institutions, the themes were remarkably similar. Thematic saturation was reached after ten participants.

This project has several strengths. First, while the prevalence of burnout is well known among emergency physicians, we believe this is the first study that examines physicians who identify as highly satisfied both in their professional and personal lives in a qualitative study. The thriving physician model points to a well-being model rather than disease-oriented model. Second, the method of a qualitive study reveals previously under-studied factors, such as social connections and the balance of introversion and extroversion. Survey-oriented studies are limited by their measurements and validated instruments used. Further, survey-oriented studies usually have very low response rates. While respondents can be compared to national norms, respondent bias is difficult to overcome.

## Conclusions

This project highlights the complexity of physician well-being and job satisfaction. Intrinsic factors, such as resilience and equanimity, interpersonal factors, such as shift-team dynamics, and extrinsic factors such as leadership culture and the EMR all contribute to influence physician job and life satisfaction. This presents a challenge for designing intervention trials, where any one intervention may not have enough impact to affect competing variables. As such, a holistic model, as the one described in this qualitative study, shows promise to inform the necessary work of physician well-being. A multi-faceted approach is necessary to cultivate those individual physician factors, while also optimizing team-dynamics, leadership, and the EMR.

## Supporting information

**S1 Appendix. Interview guide for physician thriving study.**
(DOCX)

**S2 Appendix. Transcribed interviews.**
(DOCX)

## Author Contributions

**Conceptualization:** Jesse Kase, Benjamin Doolittle.

**Data curation:** Jesse Kase, Benjamin Doolittle.

**Formal analysis:** Jesse Kase, Benjamin Doolittle.

**Investigation:** Jesse Kase, Benjamin Doolittle.

**Methodology:** Jesse Kase, Benjamin Doolittle.

**Project administration:** Jesse Kase, Benjamin Doolittle.

**Supervision:** Benjamin Doolittle.

**Validation:** Benjamin Doolittle.

**Writing – original draft:** Jesse Kase, Benjamin Doolittle.

**Writing – review & editing:** Jesse Kase, Benjamin Doolittle.

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
