## [Decision Letter · Decision Letter 0]

24 Feb 2022

PONE-D-22-01345Job and Life Satisfaction Among Emergency Physicians: a Qualitative StudyPLOS ONE

Dear Dr. Doolittle,

Thank you for submitting your manuscript to PLOS ONE. After careful consideration, we feel that it has merit but does not fully meet PLOS ONE’s publication criteria as it currently stands. Therefore, we invite you to submit a revised version of the manuscript that addresses the points raised during the review process.

We see value in this manuscript. However, the referees have raised some important concerns in regard to experimental design, presentation of the paper contribution and literature review. Thus, we ask that you carefully revise your manuscript along the lines they have suggested.

In addition, it will be important that you explicitly acknowledge the limitations and assumptions underlying your method and conclusions.

We look forward to receiving your revised manuscript.

Kind regards,

Andrea Fronzetti Colladon, Ph.D.

Academic Editor

PLOS ONE

Journal Requirements:

[No authors have competing interests.]

Additional Editor Comments:

We see value in this manuscript. However, the referees have raised some important concerns in regard to experimental design, presentation of the paper contribution and literature review. Thus, we ask that you carefully revise your manuscript along the lines they have suggested.

In addition, it will be important that you explicitly acknowledge the limitations and assumptions underlying your method and conclusions.

Reviewers' comments:

Reviewer's Responses to Questions

**Comments to the Author**

1. Is the manuscript technically sound, and do the data support the conclusions?

Reviewer #1: Yes

Reviewer #2: Yes

2. Has the statistical analysis been performed appropriately and rigorously? 

Reviewer #1: N/A

Reviewer #2: Yes

3. Have the authors made all data underlying the findings in their manuscript fully available?

Reviewer #1: Yes

Reviewer #2: No

4. Is the manuscript presented in an intelligible fashion and written in standard English?

Reviewer #1: Yes

Reviewer #2: Yes

5. Review Comments to the Author

Reviewer #1: This is a qualatiative paper regarding career and personal satisfaction in Emergency Physicians; interestingly none of the authors are emergency physicians nor work in an emergency department.

Abstract comments:

When was this study conducted? Given the impact of COVID on frontline healthcare workers (emergency physicians) this is critical information and the reader needs it for context.

23 physicians with male and Asian/South Asian skew in the population

States a model is proposed, however, there is no clear model presented in the paper

The conclusion in the abstract is a stretch. The data collection was not robutst enough to support encouraging these qualities while fostering supportive leadership and optimizine the EMR may improve satisfaction among physicians experiencing burnout. As physicians in this cohort were not studied, it is difficult to make this conclusion.

Introduction:

There are new references on burnout in emergency physicians. Consideration should be made to review some of this work and update the introcution as needed. Additionally, there are some factors in the ED that are relatively unique and impact burnout. These should be explored and discussed.

-Stehman CR, Testo Z, Gershaw RS, Kellogg AR. Burnout, Drop Out, Suicide: Physician Loss in Emergency Medicine, Part I. West J Emerg Med. 2019 May;20(3):485-494. doi: 10.5811/westjem.2019.4.40970. Epub 2019 Apr 23. Erratum in: West J Emerg Med. 2019 Aug 21;20(5):840-841. PMID: 31123550; PMCID: PMC6526882.

-West CP, Dyrbye LN, Shanafelt TD. Physician burnout: contributors, consequences and solutions. J Intern Med. 2018 Jun;283(6):516-529. doi: 10.1111/joim.12752. Epub 2018 Mar 24. PMID: 29505159.

-Ilić IM, Arandjelović MŽ, Jovanović JM, Nešić MM. Relationships of work-related psychosocial risks, stress, individual factors and burnout - Questionnaire survey among emergency physicians and nurses. Med Pr. 2017 Mar 24;68(2):167-178. English. doi: 10.13075/mp.5893.00516. Epub 2017 Mar 13. PMID: 28345677.

-Stehman CR, Clark RL, Purpura A, Kellogg AR. Wellness: Combating Burnout and Its Consequences in Emergency Medicine. West J Emerg Med. 2020 Apr 13;21(3):555-565. doi: 10.5811/westjem.2020.1.40971. PMID: 32421501; PMCID: PMC7234720.

-Watson AG, McCoy JV, Mathew J, Gundersen DA, Eisenstein RM. Impact of physician workload on burnout in the emergency department. Psychol Health Med. 2019 Apr;24(4):414-428. doi: 10.1080/13548506.2018.1539236. Epub 2018 Oct 29. PMID: 30372132.

-Truchot J, Chauvin A, Hutin A, Leredu T, Plaisance P, Yordanov Y. Burnout and satisfaction among young emergency physicians. Eur J Emerg Med. 2018 Dec;25(6):445-446. doi: 10.1097/MEJ.0000000000000526. PMID: 30379715.

Methods:

Study Design

Privately held physician management company - is this a CMG? Fee for service? Caps on number of patients per shift? Across the US? Where? Which regions? Urban vs suburban vs academic? 23 were interviewed out of how many possible? Can a response rate be calculated?

What was the rationale in excluding those with burnout? Burnout can be transient and is essentially the status quo in EM. Can EM physicians not have both burnout and high career and life satsifaction? Prior work by Shanafelt about work life satisfaction and burnout would indicate this. It follows with the "work hard, play hard" model in EM. The nature of the work may cause burnout, however, the amount of off time/free time, may not mitigate the burnout but improve career/life satisfaction.

Semi-structured interview: Who did the interviewing? What were the questions? Can you provide the interview template? Who developed the questions? Were they piloted?

Data Collection: When was this study conducted?

Demographics - what about number of children/elderly dependents in the home? The gendered expectations of women in US society greatly impacts work life balance and career satisfaction. The skew of male participants and the lack of insight into these issues negatively impacts this work. Were there any questions regarding shift number or clinical schedule? It is possible that the same individual would give different answers after 5 days off vs 5 days of working shifts.

Results:

Burnout can be transient and wax and wane, can the same be said for thriving? Can you have burnout and be thriving? Expanded horizons highlights my question above - can one be burned out and thrive? Many considered ... what percentage? I recognize that this is a qual study but a number would be helpful.

Social connections: again please clarify many

Resilience: this is complicated. Most physicians by definiation are high resilience. There are lot of challenges in schooling, standardized testing and training and those who make it through are often categorized as high resilience. Having high resilience doesn't mean that you do not have burnout. Additionally, many psychologists and social scientists would argue that resilience is determined in early childhood and that while it may be slightly modifiable, the majority of an individual's resilience has been determined by age 5-6. Please elaborate on this. Were participants specfically questioned about resilience? Would this have created bias?

Personality traits - so what? The introduction has not set up why this would be explored nor why it matters.

Balanced perspective: were participants specifically asked about emotional coping, spirituality and religion? Why are these lumped under balances perspective? Please clarify.

Above EMR as a threat, the incorrect role (roll) is used in the text.

Discussion:

Thriving physicians tended to have rich ... balance, perspective and resilience. This would have been far more novel and interesting if the same questions were asked of those with burnout. There has been a shift in the medical literature identifyling burnout as a system issue not an individual issue but this model is heavily focused on individual characteristics. Since those with burnout have been excluded, one cannot say whether or not those physicians also have balance, perspective and resilience.

Several intervention studies .... These studies show improvement in well-being and reduced burnout. Is improved well-being the same as thriving?

The paragraph on personality type feels out of place and lacks the so what or why is this important factor.

Leadership: at what level were participants asked about? ED director? CMO? CNO? Dept Chair? Corporate group leadership? There could be high variance here depending on what level of leadership is being assessed.

"transactional leadership" seems to define the leadership sytle of all CMGs and hospitals to meet CMS metrics. Please elaborate on the impact of higher level leadership and transactional leadership.

Practice setting of these physicians is critical in understanding the support from professional societies/CME statement.

Is there a typo or missing word in the sentence .. Religious involvement and spirituality .... the sentence counters itself the way it is currenlty written.

Limitations

The diversity is lacking particulalry around gender, URiM status and practice location. Please expand upon these limitations.

Please confirm this is the first study that examines physicians who identify as highly satisfied ...

This model is not novel in medicine. Please see models from Mayo and Stanford; ACEP wellness wheel; etc. How is this model different/improved?

What is the very low response rate? <1%? How low is very low?

Lack of generalizability to other practice settings (particularly academic medicine)

Conclusions:

Are all well physicians satisfied with their career and personal lives? Can you have burnout and high job satisfaction? please elaboate. How do you cultivate these indvidual factors? What are your suggested strategies?

Tables/figures

Consider a diagram of your proposed model

Table 1:

What about demographic characteristics of those who were excluded?

Satisfaction scale - is this a likert score?

Were there any individuals with disparate scores, ie high satisfied with life but moderate satisfied with career?

Reviewer #2: This is a very interesting topic and so relevant in the time of the coronavirus pandemic. The premise of identifying characteristics of EP’s that contribute to a satisfied and thriving career is refreshing after all the negative articles surrounding EP burnout. The concept of this study is truly forward-thinking.

Your authors state that their article is among few studies focusing on satisfaction in EM, yet they did not reference a study by Lall, et al “Are emergency physicians satisfied? An analysis of operational/organization factors,” (http://doi.org/10.1002/emp2.12546 ) which would add to their premise that organizational factors have the largest impact on an emergency physician’s wellbeing. Also suggest adding information from the National Academy of Medicine’s conceptual model of wellness and resilience which demonstrates that although self care is important, it is the organization that determines the bulk of an individual EP’s wellness.

The sample size in this study was small , skewed toward white males, and did not include any African American participants.

In today’s political climate, I wonder if this is a diverse enough study to be relevant for publication. The conclusions drawn from the 23 participants in this study are noteworthy, but the lack of diversity of participants is problematic.

The statistical analysis performed is satisfactory although other items to include in the one table might be: age of participants, number of years practicing emergency medicine, practice setting, etc.

The manuscript is clearly written and the themes highlighted correspond to the current literature. The statement, “Data saturation was reached after the tenth interview,” seems to be a premature conclusion. The paper lists only five common themes associated with thriving EP’s. Shanafelt listed 7 drivers of engagement in physicians so I wonder why there was a discrepancy.

This study would have significant impact if the number of physicians interviewed was much higher, with a diverse group so that more themes could be identified.

Since the themes are important to physicians, the manuscript could highlight them with the caveat that this is a study of 23 non-diverse physicians so may not be applicable to all physicians.

6. PLOS authors have the option to publish the peer review history of their article (what does this mean?). If published, this will include your full peer review and any attached files.

Reviewer #1: No

Reviewer #2: No

---

## [Author Response · Author response to Decision Letter 0]

14 Apr 2022

AUTHOR RESPONSE

Journal Requirements:

2. 

We obtained both verbal and written consent which detail in our Methods section. In addition, we note, “The interviews were recorded and transcribed. Any identifying data was removed from the transcription to preserve anonymity.” In addition, we add the number of the Yale University Institutional Review Board approved protocol, IRB # 2000022828. 

[No authors have competing interests.]

Done. We have included the line in our cover letter, “The authors have declared that no competing interests exist.”

We have uploaded the transcripts in as a “Supporting Information” file, and note this in our cover letter. 

We have double checked that this is the case.

We added the line, “Both recorded verbal and written consent was obtained at the beginning of the interview.” In addition, we add the number of the Yale University Institutional Review Board approved protocol, IRB # 2000022828. 

Additional Editor Comments:

We see value in this manuscript. However, the referees have raised some important concerns in regard to experimental design, presentation of the paper contribution and literature review. Thus, we ask that you carefully revise your manuscript along the lines they have suggested.

In addition, it will be important that you explicitly acknowledge the limitations and assumptions underlying your method and conclusions.

Reviewers' comments:

Reviewer's Responses to Questions

Comments to the Author

1. Is the manuscript technically sound, and do the data support the conclusions?

Reviewer #1: Yes

Reviewer #2: Yes

2. Has the statistical analysis been performed appropriately and rigorously? 

Reviewer #1: N/A

Reviewer #2: Yes

3. Have the authors made all data underlying the findings in their manuscript fully available?

Reviewer #1: Yes

Reviewer #2: No

4. Is the manuscript presented in an intelligible fashion and written in standard English?

Reviewer #1: Yes

Reviewer #2: Yes

 

5. Review Comments to the Author

Reviewer #1: This is a qualitative paper regarding career and personal satisfaction in Emergency Physicians; interestingly none of the authors are emergency physicians nor work in an emergency department.

To both reviewers, thank you for your overall support of our manuscript and for your thoughtful comments. We recognize that these are busy times. Given the level of detailed, insightful comments, you certainly spent significant time with the manuscript. Please know how deeply we appreciate your efforts. We have incorporated your suggestions throughout the manuscript and respond in detail to your comments below.

In particular, the other papers that you propose are excellent. Initially, we struggled with how much detail to include. We have incorporated those papers throughout the manuscript. With your suggestions, the manuscript is much stronger. Many thanks. 

Of note, the senior author worked for 16 years – the night shift – in a community hospital emergency department. In part, this project evolved from a senior leader at the private emergency department group approaching me to explore issues of burnout and well-being. 

Abstract comments:

When was this study conducted? Given the impact of COVID on frontline healthcare workers (emergency physicians) this is critical information and the reader needs it for context.

23 physicians with male and Asian/South Asian skew in the population

States a model is proposed, however, there is no clear model presented in the paper

The conclusion in the abstract is a stretch. The data collection was not robust enough to support encouraging these qualities while fostering supportive leadership and optimizing the EMR may improve satisfaction among physicians experiencing burnout. As physicians in this cohort were not studied, it is difficult to make this conclusion.

We have added the dates when the interviews took place (June-September 2020) and are mindful that the project took place at varying stages of the pandemic. Although, at the hospitals we studied, the COVID-19 pandemic was in a temporary lull. 

The demographics in our sample were similar to a large 2020 data set of Emergency Physicians: https://reader.elsevier.com/reader/sd/pii/S0196064420305011?token=84857CF28F19E35C6A5089F05C41DA0F98ADB6FDB30FC5974B8008292FFE9BF4C173E56DE36B4C0D8A581B431CE2344D&originRegion=us-east-1&originCreation=20220331201623

This study showed a median age of 50, with 72% male. While ethnicity data was not included in that large study of ED physicians, national data for all physicians show 56.2% Caucasian, 17.1% Asian, 5.8% Hispanic, 5.0% Black. https://www.aamc.org/data-reports/workforce/interactive-data/figure-18-percentage-all-active-physicians-race/ethnicity-2018. Another large study published in the literature (referenced by reviewer #2, https://doi.org/10.1002/emp2.12546) 76% male, average age of 50.82, and 84% Caucasian. 

We believe that our representation, in general, represents that demographics of the physician workforce. We have referenced these papers in our results section. 

A cohort of 23 participants for a qualitative review is sufficient, especially when thematic saturation was reached after 10 participants. Many qualitative studies published in PLOSONE had a similar number of participants. Twenty one primary care doctors were studied in a suicide prevention project (https://doi.org/10.1371/journal.pone.0242540), and seventeen were studied in a project about supportive roles of Black and Latinx health care workers during the COVID-19 pandemic (https://doi.org/10.1371/journal.pone.0262606). 

Since qualitative studies are designed to draw out naturally occurring themes, our recommendations are based on this input. EMR and leadership issues have been cited in other projects, which we now include in the discussion at the recommendation of reviewer #2 (https://doi.org/10.1016/j.mayocp.2016.10.004). This qualitative study affirms the importance of optimizing the EMR, workflow, and leadership.

Since one of the identified stressors was the lack of support and challenges around leadership and administration, we believe it is relevant to comment about this in the conclusion. We have incorporated the paper suggested by reviewer #2 and included this in the conclusion, “Shanafelt et al describes seven drivers that leaders need to address to improve physician engagement: resources, workload, control/flexibility, work-life integration, social support, meaning in work, organizational culture. This model affirms the findings derived from the lived-experience of emergency physicians in this qualitative study. In particular, optimizing the work environment with appropriate resources and workload as well as an emphasis work-place culture through responsive leadership and social support were key drivers in our model.” 

Introduction:

There are new references on burnout in emergency physicians. Consideration should be made to review some of this work and update the introduction as needed. Additionally, there are some factors in the ED that are relatively unique and impact burnout. These should be explored and discussed.

-Stehman CR, Testo Z, Gershaw RS, Kellogg AR. Burnout, Drop Out, Suicide: Physician Loss in Emergency Medicine, Part I. West J Emerg Med. 2019 May;20(3):485-494. doi: 10.5811/westjem.2019.4.40970. Epub 2019 Apr 23. Erratum in: West J Emerg Med. 2019 Aug 21;20(5):840-841. PMID: 31123550; PMCID: PMC6526882.

-West CP, Dyrbye LN, Shanafelt TD. Physician burnout: contributors, consequences and solutions. J Intern Med. 2018 Jun;283(6):516-529. doi: 10.1111/joim.12752. Epub 2018 Mar 24. PMID: 29505159.

-Ilić IM, Arandjelović MŽ, Jovanović JM, Nešić MM. Relationships of work-related psychosocial risks, stress, individual factors and burnout - Questionnaire survey among emergency physicians and nurses. Med Pr. 2017 Mar 24;68(2):167-178. English. doi: 10.13075/mp.5893.00516. Epub 2017 Mar 13. PMID: 28345677.

-Stehman CR, Clark RL, Purpura A, Kellogg AR. Wellness: Combating Burnout and Its Consequences in Emergency Medicine. West J Emerg Med. 2020 Apr 13;21(3):555-565. doi: 10.5811/westjem.2020.1.40971. PMID: 32421501; PMCID: PMC7234720.

-Watson AG, McCoy JV, Mathew J, Gundersen DA, Eisenstein RM. Impact of physician workload on burnout in the emergency department. Psychol Health Med. 2019 Apr;24(4):414-428. doi: 10.1080/13548506.2018.1539236. Epub 2018 Oct 29. PMID: 30372132.

-Truchot J, Chauvin A, Hutin A, Leredu T, Plaisance P, Yordanov Y. Burnout and satisfaction among young emergency physicians. Eur J Emerg Med. 2018 Dec;25(6):445-446. doi: 10.1097/MEJ.0000000000000526. PMID: 30379715.

Thank you. We have expanded the introduction to include several of these citations. We thought that the West, Ilić, and first Stehman articles were the most relevant for the introduction, and we used the Truchot article to support the conclusion. 

Methods:

Study Design

Privately held physician management company - is this a CMG? Fee for service? Caps on number of patients per shift? Across the US? Where? Which regions? Urban vs suburban vs academic? 23 were interviewed out of how many possible? Can a response rate be calculated?

We have added the line, “… contract management group that staffs several community-based hospitals in the western United States.” We felt it necessary to protect the identity of the CMG. 

What was the rationale in excluding those with burnout? Burnout can be transient and is essentially the status quo in EM. Can EM physicians not have both burnout and high career and life satisfaction? Prior work by Shanafelt about work life satisfaction and burnout would indicate this. It follows with the "work hard, play hard" model in EM. The nature of the work may cause burnout, however, the amount of off time/free time, may not mitigate the burnout but improve career/life satisfaction.

The rationale for excluding those with burnout was to explore a model of thriving. Burnout – and its associated factors – has been well described in the literature. However, emergency physicians who are thriving – the positive deviants – are not well described. Who is happy in their life and their career? There are very few projects among physicians which explore this concept. While we recognize that physicians may be burned out at work and have high life-satisfaction, we chose to explore a model of holistic well-being, since this is not well-studied in the literature. We have clarified this in the Introduction with the line, “Given that burnout is well-described in the literature, this project explored a model of physician thriving, as in, those who are not experiencing burnout and are satisfied both in their work and their life.”

Your point about the “work hard, play hard” spirit of EM is a good one. In the Discussion section, we explore this idea, “While it is conceptually possible for an emergency physician to experience burnout at work and high satisfaction at home, or vice-versa, this project explored a holistic model of both high job and life satisfaction. The Institute of Healthcare Improvement supplements their Triple Aim of healthcare, quality, efficiency, and patient experience, with a fourth aim, physician work satisfaction. We chose to explore a model of high job and life satisfaction. Our model describes the importance of both work and life satisfaction. Several quantitative projects affirm the importance of a balanced work and home life among physicians.

Of note, we believe that the mixed-methods nature of this study is one of its strengths. Many questionnaire studies – even those by Shanafelt et al – are plagued by low response rate. Many qualitative studies do not discriminate among those who are burned out or not. 

Semi-structured interview: Who did the interviewing? What were the questions? Can you provide the interview template? Who developed the questions? Were they piloted?

Data Collection: When was this study conducted?

Demographics - what about number of children/elderly dependents in the home? The gendered expectations of women in US society greatly impacts work life balance and career satisfaction. The skew of male participants and the lack of insight into these issues negatively impacts this work. Were there any questions regarding shift number or clinical schedule? It is possible that the same individual would give different answers after 5 days off vs 5 days of working shifts.

The first author conducted the interviews, which we now note in the Methods section. The quantitative measures (burnout and life satisfaction) are validated instruments which we cite in the Methods section. The question template was derived from the literature on well-being and burnout, the authors’ own conceptual framework, and focused on broad themes of work culture, social support, emotional coping, and other strategies. The questions are shared in the Appendix. The same template has been used in a study among primary care physicians, which is cited (#10). The open-ended nature of the questions allowed for broad responses. Of note, participants did not voluntarily share the benefits of working 5 days on and 5 days off, nor any other specific schedule arrangement. 

Results:

Burnout can be transient and wax and wane, can the same be said for thriving? Can you have burnout and be thriving? Expanded horizons highlights my question above - can one be burned out and thrive? Many considered ... what percentage? I recognize that this is a qual study but a number would be helpful.

We recognize that we are exploring new territory here, and much is unknown. You raise a very provocative question about being burned out and still thriving, since these are slightly different constructs. As you likely know, Shanafelt’s work shows that emergency physicians are among the most burned out AND the most satisfied with work-life balance (https://doi.org/10.1016/j.mayocp.2015.08.023). 

In an attempt to have a well-defined model, we chose to focus on those physicians who were satisfied in both career and life. Since we used snow-ball sampling where satisfied physicians referred other satisfied physicians, we do not have aggregate scores of burnout, work and life satisfaction among all the emergency physicians in the CMG. 

Social connections: again please clarify many

Resilience: this is complicated. Most physicians by definition are high resilience. There are lot of challenges in schooling, standardized testing and training and those who make it through are often categorized as high resilience. Having high resilience doesn't mean that you do not have burnout. Additionally, many psychologists and social scientists would argue that resilience is determined in early childhood and that while it may be slightly modifiable, the majority of an individual's resilience has been determined by age 5-6. Please elaborate on this. Were participants specfically questioned about resilience? Would this have created bias?

Personality traits - so what? The introduction has not set up why this would be explored nor why it matters.

Balanced perspective: were participants specifically asked about emotional coping, spirituality and religion? Why are these lumped under balances perspective? Please clarify.

Above EMR as a threat, the incorrect role (roll) is used in the text. 

You raise several points here that reflect the nature of qualitative studies from semi-structured interviews. 

We are not testing hypotheses, such as whether resilience correlates with burnout. Instead, we are drawing conclusions based on the naturally occurring themes among physicians who are not burned out and identify with high work and life satisfaction. This leads to findings that make sense, such as the importance of resilience. Other findings may be surprising, such as the perspective of expanded horizons or spirituality, which are not often explored in large quantitative studies of physician burnout. 

In the semi-structured interview guide, several open-ended questions are asked to elicit coping strategies such as, “What traits do you have that help you be successful in your job?”, “Is there anything about your job that has helped you thrive?” “What helps the most to have a successful career?” The art of qualitative studies is not to ask specific leading questions – about resilience for example – but rather to follow up and expand on those themes when the participant offers them. 

Thank you for picking up the typo. The correct “role” (not roll) is used. 

Discussion:

Thriving physicians tended to have rich ... balance, perspective and resilience. This would have been far more novel and interesting if the same questions were asked of those with burnout. There has been a shift in the medical literature identifying burnout as a system issue not an individual issue but this model is heavily focused on individual characteristics. Since those with burnout have been excluded, one cannot say whether or not those physicians also have balance, perspective and resilience.

Several intervention studies .... These studies show improvement in well-being and reduced burnout. Is improved well-being the same as thriving?

The paragraph on personality type feels out of place and lacks the so what or why is this important factor.

Leadership: at what level were participants asked about? ED director? CMO? CNO? Dept Chair? Corporate group leadership? There could be high variance here depending on what level of leadership is being assessed.

"transactional leadership" seems to define the leadership style of all CMGs and hospitals to meet CMS metrics. Please elaborate on the impact of higher level leadership and transactional leadership.

Practice setting of these physicians is critical in understanding the support from professional societies/CME statement.

There are several important questions here that we share with the reviewer. Since the purpose of a qualitative study is to explore novel themes derived from participants’ lived-experience, by definition, the granular quality of these issues may not be well-elucidated. Quantitative, survey-type, studies are also limited in this way. We believe that the strength of our mixed-methods strategy is that thriving physicians describe both intrinsic and extrinsic factors that influence their job and life satisfaction. This takes a holistic view of the emergency physician, which we believe is an important contribution that complements the work of Shanafelt, Lall, and others. The level of leadership – ED director, CMO, etc. – may vary among institution and participant perception. The signal from our participants is that leadership matters. Thus, we felt it important to include data about transactional leadership. Certainly, further study about leadership, ED setting, personality traits, team dynamics, etc. is merited. 

Is there a typo or missing word in the sentence .. Religious involvement and spirituality .... the sentence counters itself the way it is currently written.

We have clarified the point about religious involvement to read, “Of note, our participants did not cite organized religious involvement and other formal social clubs as important to well-being, which has been noted in broader population based studies.”

Limitations

The diversity is lacking particularly around gender, URiM status and practice location. Please expand upon these limitations.

The diversity of our sample size generally reflects the diversity of emergency physicians, as noted in several studies that I cite above. We now note this in the limitations section with the line, “While most physicians were male and Caucasian, our sample reflects similar demographics of larger, quantitative studies.” 

Please confirm this is the first study that examines physicians who identify as highly satisfied ...

This model is not novel in medicine. Please see models from Mayo and Stanford; ACEP wellness wheel; etc. How is this model different/improved?

What is the very low response rate? <1%? How low is very low?

Lack of generalizability to other practice settings (particularly academic medicine)

Conclusions:

Are all well physicians satisfied with their career and personal lives? Can you have burnout and high job satisfaction? please elaborate. How do you cultivate these individual factors? What are your suggested strategies?

While we believe this is the first study of this type among emergency physicians, we soften this claim by the line in the Discussion, “…we believe this is the first study that examines physicians who identify as highly satisfied both in their professional and personal lives in a qualitative study.” 

The models derived by Mayo, Lall, Shanafelt and others are derived from large survey-based studies. Our study, to our knowledge, is the first among emergency physicians to employ qualitative studies to explore physicians who are satisfied both in work and life. The novelty is our mixed-methods approach: validated instruments combined with open-ended interviews. This allowed for some interesting findings. In particular, physicians described themselves as both introverted and extroverted. This is a difficult construct to capture in a validated survey, but precisely the sort of nuanced finding that can be elicited from interviews. 

Since both reviewers note that it is conceptually possible to be both burned out at work, but have high life satisfaction, and vice-versa, we include in the limitations section, “Of note, this project was designed to explore a model of thriving through the lived-experience of front-line emergency physicians. It was not designed to explore factors related to burnout.” 

We also note in our limitations section that these physicians served several community-based hospitals in the western United States, but worked for the same CMG. We have further added in this revision, “these findings may be not be generalizable to all emergency physicians.”

Tables/figures

Consider a diagram of your proposed model

Table 1:

What about demographic characteristics of those who were excluded?

Satisfaction scale - is this a likert score?

Yes, the satisfaction scale is a validated Likert scale, widely used in the literature. We cite this in the Methods section. Since recruitment was self-referential and through the snowball effect, only two physicians met criteria for burnout and were not included in the analysis, since this project was to build a model of physician thriving. 

Were there any individuals with disparate scores, ie high satisfied with life but moderate satisfied with career?

No, as physicians were recruited through self-reference and snow-ball sampling, only physicians who met criteria for high life and work satisfaction, and were not experiencing burnout were included. This was to derive a model for thriving. 

Reviewer #2: This is a very interesting topic and so relevant in the time of the coronavirus pandemic. The premise of identifying characteristics of EP’s that contribute to a satisfied and thriving career is refreshing after all the negative articles surrounding EP burnout. The concept of this study is truly forward-thinking.

Thank you!

Your authors state that their article is among few studies focusing on satisfaction in EM, yet they did not reference a study by Lall, et al “Are emergency physicians satisfied? An analysis of operational/organization factors,” (http://doi.org/10.1002/emp2.12546) which would add to their premise that organizational factors have the largest impact on an emergency physician’s wellbeing. Also suggest adding information from the National Academy of Medicine’s conceptual model of wellness and resilience which demonstrates that although self care is important, it is the organization that determines the bulk of an individual EP’s wellness.

The sample size in this study was small , skewed toward white males, and did not include any African American participants.

In today’s political climate, I wonder if this is a diverse enough study to be relevant for publication. The conclusions drawn from the 23 participants in this study are noteworthy, but the lack of diversity of participants is problematic.

The Lall article is excellent, and affirms our findings. Although, we recognize that Lall takes an institutional approach, based on a survey, rather than exploring personality traits and work environment in qualitative study. We believe both models complement each other. We incorporate this paper into our discussion, “In addition, Lall et al derived a model of job satisfaction from a survey-based study that identified the importance of reducing burnout, opportunities for non-clinical opportunities, optimizing work-conditions, family time, and the electronic health record, and reducing discrimination. Our qualitative study adds validity to the Shanafelt and Lall models. In addition, our study explores intrinsic features such as resilience, balanced perspective, and being both introverted and extroverted.”

Reviewer #1 raised the same issue about diversity and study sample. I include my response here as well:

The demographics in our sample were similar to a large 2020 data set of Emergency Physicians: https://reader.elsevier.com/reader/sd/pii/S0196064420305011?token=84857CF28F19E35C6A5089F05C41DA0F98ADB6FDB30FC5974B8008292FFE9BF4C173E56DE36B4C0D8A581B431CE2344D&originRegion=us-east-1&originCreation=20220331201623

This study showed a median age of 50, with 72% male. While ethnicity data was not included in that large study of ED physicians, national data for all physicians show 56.2% Caucasian, 17.1% Asian, 5.8% Hispanic, 5.0% Black. https://www.aamc.org/data-reports/workforce/interactive-data/figure-18-percentage-all-active-physicians-race/ethnicity-2018. Another large study published in the literature (referenced by reviewer #2, https://doi.org/10.1002/emp2.12546) 76% male, average age of 50.82, and 84% Caucasian. 

We believe that our representation, in general, represents that demographics of the physician workforce. We have referenced these papers in our results section. 

A cohort of 23 participants for a qualitative review is sufficient, especially when thematic saturation was reached after 10 participants. Many qualitative studies published in PLOSONE had a similar number of participants. Twenty one primary care doctors were studied in a suicide prevention project (https://doi.org/10.1371/journal.pone.0242540), and seventeen were studied in a project about supportive roles of Black and Latinx health care workers during the COVID-19 pandemic (https://doi.org/10.1371/journal.pone.0262606). 

Since this was a study to explore thriving only, and not burnout, and recruitment was self-selected and via snowball sampling, only two physicians were screened, met criteria for burnout, and thus were not included in the final analysis. 

The statistical analysis performed is satisfactory although other items to include in the one table might be: age of participants, number of years practicing emergency medicine, practice setting, etc.

The manuscript is clearly written and the themes highlighted correspond to the current literature. The statement, “Data saturation was reached after the tenth interview,” seems to be a premature conclusion. The paper lists only five common themes associated with thriving EP’s. Shanafelt listed 7 drivers of engagement in physicians so I wonder why there was a discrepancy.

As we reviewed the transcripts in an iterative fashion, the output of themes was stable after the tenth interview. From ten to twenty-three, no new themes emerged, hence the statement about data saturation. 

Shanafelt’s paper (https://doi.org/10.1016/j.mayocp.2016.10.004) describes nine drivers of engagement – resources, workload, control/flexibility, work-life integration, social support, meaning in work, organizational culture – that pertain to institutional interventions. This model is derived from his long-standing leadership in the field and his extensive surveys. Our model is more limited in scope and is derived from the lived experienced of front-line emergency physicians. Although different, his model does resonate with our five themes of expanded horizons, social connections, resilience, balanced perspective, qualities of both introversion and extroversion. 

To incorporate this important paper, we have included the following in the discussion, “This model affirms the findings by Shanafelt et al who describes nine drivers that leaders need to address to improve physician engagement: resources, workload, control/flexibility, work-life integration, social support, meaning in work, organizational culture. This model affirms the findings derived from the lived-experience of emergency physicians in this qualitative study. In particular, optimizing the work environment with appropriate resources and workload as well as an emphasis work-place culture through responsive leadership and social support were key drivers in our model.”

This study would have significant impact if the number of physicians interviewed was much higher, with a diverse group so that more themes could be identified.

Since the themes are important to physicians, the manuscript could highlight them with the caveat that this is a study of 23 non-diverse physicians so may not be applicable to all physicians.

As we note above, 23 physicians is an appropriate number for a qualitative study, especially when thematic saturation was reached after 10 participants. Several qualitative studies exploring well-being and burnout among health care providers had similar numbers. A study among emergency physician residents employed four focus groups, and did not include the total number of participants (https://doi.org/10.1080/10401334.2021.1875833); a study among physicians in rural Canada had 14 participants (https://doi.org/10.1186/s12913-021-06899-y); a study among primary care physicians’ satisfaction with telehealth during the COVID-19 pandemic had 15 participants. 

The question of diversity is valid. However, as noted above, most emergency physicians are male. The ethnic diversity of our cohort is similar to the overall diversity of physicians in the United States. We note three papers (see above) where the diversity of our sample was similar to other published studies. We now cite these in the Results and Discussion section. We also note in the Limitations section, “While most physicians were male and Caucasian, our sample reflects similar demographics of larger, quantitative studies.” 

Many thanks for your thoughtful review and support of this project. You both raised excellent points. We hope that we have responded thoughtfully and completely to each of them. We have incorporated nearly all the articles you cite into the manuscript, softened some of our claims, and expanded both the introduction and discussion. All the best….

---

## [Decision Letter · Decision Letter 1]

29 Apr 2022

PONE-D-22-01345R1Job and Life Satisfaction Among Emergency Physicians: a Qualitative StudyPLOS ONE

Dear Dr. Doolittle,

Thank you for submitting your manuscript to PLOS ONE. After careful consideration, we feel that it has merit but does not fully meet PLOS ONE’s publication criteria as it currently stands. Therefore, we invite you to submit a revised version of the manuscript that addresses the points raised during the review process.

We look forward to receiving your revised manuscript.

Kind regards,

Andrea Fronzetti Colladon, Ph.D.

Academic Editor

PLOS ONE

Reviewers' comments:

Reviewer's Responses to Questions

**Comments to the Author**

1. If the authors have adequately addressed your comments raised in a previous round of review and you feel that this manuscript is now acceptable for publication, you may indicate that here to bypass the “Comments to the Author” section, enter your conflict of interest statement in the “Confidential to Editor” section, and submit your "Accept" recommendation.

Reviewer #1: (No Response)

2. Is the manuscript technically sound, and do the data support the conclusions?

Reviewer #1: Yes

3. Has the statistical analysis been performed appropriately and rigorously? 

Reviewer #1: Yes

4. Have the authors made all data underlying the findings in their manuscript fully available?

Reviewer #1: No

5. Is the manuscript presented in an intelligible fashion and written in standard English?

Reviewer #1: Yes

6. Review Comments to the Author

Reviewer #1: Thank you for your submitted revision of this manuscript.

There are still some underlying methodological issues that have not been adequately addressed.

In reviewing the transcripts, I have some additional questions/concerns.

It seems that not everyone was asked all of the same questions, particularly JC.

Given the sample size, there seem to be a relatively high number of medical directors/other leadership in this cohort. Please address this in the manuscript particularly considering that leadership/local hospital admin was a threat.

While age is described in table 1, perhaps it would be more meaningful to use years in practice. It seems that total years in practice was not asked, only years in the current job. Is this correct? If so, is this a potential limitation or co-founder? Years in current job should also be added to Table 1. There are some with many years in the same job but many with relatively few years in the current job.

Did this cohort include a pediatrician and IM or FM physician? This should be clarified.

Were participants asked about admin hours/month? Clearly there are some non-clinical work hours for the medical directors/admin folks and likely some for the purely clinical folks too. There seemed to be inconsistencies in asking about the number of hours worked.

The sample seemed to include several individuals that would meet criteria for burnout and/or scored 6 or lower on the satisfaction scale. By your definition (>7), shouldn't these people have been excluded? There were more than 2 individuals who met one of these criteria.

Abstract

Please add the study dates to the abstract

Methods

Appendix A - where is this? It was not provided in the materials to review.

Additionally, these questions were a "standardized list". Please clarify. Where was the list derived from? How were these questions decided upon?

My prior comment was misunderstood. As we know that burnout can be transient, it would seem important to have ascertained when these interviews occurred. If most interviews occurred after someone had one day off vs multiple days off vs just worked multiple days in a row, the answers provided in the qualitative survey may have been different among the same individual. There were mention of external (home) stressors by several participants and this should also be noted.

Results

Please provide rates for gender and ethnicity in the body of the paper as they are presented in the abstract.

The first paragraph is confusing and contradicts itself, please revise. "None of the physicians met criteria for burnout." "Two physicians contacted by the interviewer met criteria for burnout ..."

Unclear if their pre-screen survey was negative and then responses changed at the time of the interview.

Based on the methodology, should those with life satisfaction or career satisfaction less than 7 have been excluded?

Strengths/Limitations

It should be mentioned that this survey was conducted during COVID. The authors mention in their response letter that rates in the West were low at that time, however, vaccination wasn't yet available and there were still so many clinical unknowns that it must be stated that COVID likely impacted satisfaction. It is the elephant in the room and must be addressed in some way.

As this was a community-based study, it should be specified that these findings may not be generalizable to academic emergency physicians.

Please specify the "diversity of participants" further. In years of practice? Practice location? Seniority? Leadership?

Consider creating a visual representation of the "thriving physician model" from your results.

7. PLOS authors have the option to publish the peer review history of their article (what does this mean?). If published, this will include your full peer review and any attached files.

Reviewer #1: No

---

## [Author Response · Author response to Decision Letter 1]

12 Jun 2022

RESPONSE TO REVIEWERS

Reviewers' comments:

Reviewer's Responses to Questions

Comments to the Author

1. If the authors have adequately addressed your comments raised in a previous round of review and you feel that this manuscript is now acceptable for publication, you may indicate that here to bypass the “Comments to the Author” section, enter your conflict of interest statement in the “Confidential to Editor” section, and submit your "Accept" recommendation.

Reviewer #1: (No Response)

2. Is the manuscript technically sound, and do the data support the conclusions?

Reviewer #1: Yes

3. Has the statistical analysis been performed appropriately and rigorously?

Reviewer #1: Yes

4. Have the authors made all data underlying the findings in their manuscript fully available?

Reviewer #1: No

Please note that we have uploaded our interview transcripts.

5. Is the manuscript presented in an intelligible fashion and written in standard English?

Reviewer #1: Yes

6. Review Comments to the Author

Reviewer #1: Thank you for your submitted revision of this manuscript.

Thank you for your ongoing support of our project. We appreciate the conscientiousness of your comments and your careful eye. We have made several edits throughout, which we believe makes the manuscript much stronger. Many thanks. 

There are still some underlying methodological issues that have not been adequately addressed.

In reviewing the transcripts, I have some additional questions/concerns.

It seems that not everyone was asked all of the same questions, particularly JC.

The interviews were semi-structured. As you see in the transcripts, the questions are open-ended and allow for non-biased responses. While nearly the same questions were asked of all participants, there may be variation. JC was the first interview and several important themes were captured. As we refined the themes, based on emergent ideas, further interviews delved into those themes more deeply. This is the process of grounded theory in qualitative studies. 

Given the sample size, there seem to be a relatively high number of medical directors/other leadership in this cohort. Please address this in the manuscript particularly considering that leadership/local hospital admin was a threat.

This is a good point. Of the 23, eight had administrative duties. We clarify this in the results with the line, “Eight (35%) of the participants held leadership positions: regional director (1), medical director (5), assistant medical director (2).” 

In the limitations section, we add the line, “Eight of the participants held leadership positions. This may represent a confounding variable. They may experience extra pressure in their leadership roles or less pressure since their administrative duties lessen their direct-facing patient care. However, their responses resonated with non-leaders with the one difference being that the did not regard leadership as a threat.” 

While age is described in table 1, perhaps it would be more meaningful to use years in practice. It seems that total years in practice was not asked, only years in the current job. Is this correct? If so, is this a potential limitation or co-founder? Years in current job should also be added to Table 1. There are some with many years in the same job but many with relatively few years in the current job.

While we did ask the participant’s age and total number of years in the present job, we did not ask about total years as an ED physician. We have included this in Table 1. With an average age of 45, most participants were mid-career. While length of time as an ED physician may be helpful for quantitative, survey-based studies, for this qualitative study to explore physician thriving, a broad, representative sample of ED physicians that generally reflect the demographics of ED physicians nationally is helpful here.

Did this cohort include a pediatrician and IM or FM physician? This should be clarified.

All were board certified ED physicians. We now note this in the results section. 

Were participants asked about admin hours/month? Clearly there are some non-clinical work hours for the medical directors/admin folks and likely some for the purely clinical folks too. There seemed to be inconsistencies in asking about the number of hours worked.

In several quantitative studies, increased work hours is associated with burnout. However, in this qualitative study, we explored broad themes of personality types, psycho-social support, and other factors in an open-ended, semi-structured fashion. We were not specifically controlling for work hours. In these interviews, work hours specifically did not emerge as a specific factor, but rather, being socially connected with significant outside interests did. This is the particular strength of qualitative studies and this project in particular. The issue regarding thriving may not be work hours, but rather the quality of time within and outside of work. In a similar study, work hours also did not emerge as a notable theme, but social connection did (JGIM 2021 Dec;36(12):3759-3765. doi: 10.1007/s11606-021-06883-6.) 

The sample seemed to include several individuals that would meet criteria for burnout and/or scored 6 or lower on the satisfaction scale. By your definition (>7), shouldn't these people have been excluded? There were more than 2 individuals who met one of these criteria.

While these interviews were included in the transcripts, they were excluded from analysis. 

Abstract

Please add the study dates to the abstract

Done. We have added, “… from June-September 2020.”

Methods

Appendix A - where is this? It was not provided in the materials to review.

I uploaded this but apparently it was not in the final pdf. I will make sure it is included this time around. 

Additionally, these questions were a "standardized list". Please clarify. Where was the list derived from? How were these questions decided upon?

My prior comment was misunderstood. As we know that burnout can be transient, it would seem important to have ascertained when these interviews occurred. If most interviews occurred after someone had one day off vs multiple days off vs just worked multiple days in a row, the answers provided in the qualitative survey may have been different among the same individual. There were mention of external (home) stressors by several participants and this should also be noted.

In general, burnout does not necessarily change dramatically based on a few days off, but tends to be stable over time. Since this project was designed to explore broad themes of physician thriving – and not burnout – the specifics of whether a participant had a day off seems not so relevant here. The questions are designed to be broad, “Overall, how satisfied are you….” “Share with us your views about work-life balance….”

Physician thriving is a relatively new concept in the literature. While burnout has been well-studied – indeed, many of us are burned out from burnout – what works has not been well understood or well-described. What makes a physician thrive? We thought it appropriate to ask physicians who were satisfied with life and satisfied with their career (two validated instruments) and also were not burned out (using another validated instrument). The initial questions were derived from review of the literature, namely the work of Tait Shanafelt and others, of which much is theoretical and has not been explored in qualitative studies. Also, as the interviews are semi-structured, the questions are meant to serve as a starting point for a reflective conversation. We believe this is an important strength of the project. 

Results

Please provide rates for gender and ethnicity in the body of the paper as they are presented in the abstract.

 Done.

The first paragraph is confusing and contradicts itself, please revise. "None of the physicians met criteria for burnout." "Two physicians contacted by the interviewer met criteria for burnout ..."

Unclear if their pre-screen survey was negative and then responses changed at the time of the interview.

We have clarified this to read, “There were 25 interviews conducted with emergency physicians across 10 hospitals [Table 1]. Two physicians contacted by the interviewer met criteria for burnout and were not included in the study, leaving 23 for data analysis.” 

Based on the methodology, should those with life satisfaction or career satisfaction less than 7 have been excluded?

This is correct. Two did not meet our inclusion criteria and were excluded from analysis. 

Strengths/Limitations

It should be mentioned that this survey was conducted during COVID. The authors mention in their response letter that rates in the West were low at that time, however, vaccination wasn't yet available and there were still so many clinical unknowns that it must be stated that COVID likely impacted satisfaction. It is the elephant in the room and must be addressed in some way.

As this was a community-based study, it should be specified that these findings may not be generalizable to academic emergency physicians.

Please specify the "diversity of participants" further. In years of practice? Practice location? Seniority? Leadership?

In the discussion section, we address the COVID-19 pandemic, “These interviews were conducted during a relative trough in the COVID-19 pandemic at many of the hospitals. We believe this makes these results compelling since these physicians met criteria for thriving and were not burned out during a pandemic. These emergent themes – resilience, social connection, outside interests, with personality types of introversion and extroversion – were associated with career and life satisfaction during a particularly stressful period for the healthcare system.” 

In our limitations section, we did have a line, “these findings may not be generalizable to all emergency physicians.” Now, we add the clause, “especially those in academic medicine as this was a community-based study.” 

In our results section, we do have the line, “Ethnicity and gender were similar to other large studies of emergency physicians [17-19].” We affirm this again in our discussion. Now, in the limitations section, we discuss that 8 of the 23 participants held leadership positions. 

Consider creating a visual representation of the "thriving physician model" from your results.

We have done so… It looks pretty good!

7. PLOS authors have the option to publish the peer review history of their article (what does this mean?). If published, this will include your full peer review and any attached files.

Do you want your identity to be public for this peer review? For information about this choice, including consent withdrawal, please see our Privacy Policy.

Reviewer #1: No

---

## [Decision Letter · Decision Letter 2]

13 Jul 2022

PONE-D-22-01345R2Job and Life Satisfaction Among Emergency Physicians: a Qualitative StudyPLOS ONE

Dear Dr. Doolittle,

Thank you for submitting your manuscript to PLOS ONE. After careful consideration, we feel that it has merit but does not fully meet PLOS ONE’s publication criteria as it currently stands. Therefore, we invite you to submit a revised version of the manuscript that addresses the points raised during the review process.

ACADEMIC EDITOR: Please respond to all suggested revisions and inquiries by the reviewersIn particular, please carefully address Reviewer #3 comments about the reporting of the results. I agree that the results section would benefit from more thorough reporting of the details and characteristics of each theme with additional supporting quotes where possible. For example, for the theme "Electronic Medical Record as a threat" what specific aspects of the EMR were "dreaded"? Did anyone feel it was not an issue? Currently this theme includes a one sentence descriptor and one sentence quote. Providing more detail  is needed to fully characterize the themes that emerged from the interviews, will help readers to better understand the findings, and may contribute to potential solutions. Please submit your revised manuscript by Aug 27 2022 11:59PM. If you will need more time than this to complete your revisions, please reply to this message or contact the journal office at plosone@plos.org. Please include the following items when submitting your revised manuscript:A rebuttal letter that responds to each point raised by the academic editor and reviewer(s). You should upload this letter as a separate file labeled 'Response to Reviewers'.A marked-up copy of your manuscript that highlights changes made to the original version. You should upload this as a separate file labeled 'Revised Manuscript with Track Changes'.An unmarked version of your revised paper without tracked changes. You should upload this as a separate file labeled 'Manuscript'.

We look forward to receiving your revised manuscript.

Kind regards,

Niklas Bobrovitz

Academic Editor

PLOS ONE

Reviewers' comments:

Reviewer's Responses to Questions

**Comments to the Author**

1. If the authors have adequately addressed your comments raised in a previous round of review and you feel that this manuscript is now acceptable for publication, you may indicate that here to bypass the “Comments to the Author” section, enter your conflict of interest statement in the “Confidential to Editor” section, and submit your "Accept" recommendation.

Reviewer #1: (No Response)

Reviewer #3: (No Response)

2. Is the manuscript technically sound, and do the data support the conclusions?

Reviewer #1: Yes

Reviewer #3: Partly

3. Has the statistical analysis been performed appropriately and rigorously? 

Reviewer #1: Yes

Reviewer #3: Yes

4. Have the authors made all data underlying the findings in their manuscript fully available?

Reviewer #1: Yes

Reviewer #3: No

5. Is the manuscript presented in an intelligible fashion and written in standard English?

Reviewer #1: Yes

Reviewer #3: Yes

6. Review Comments to the Author

Reviewer #1: Thank you for your additional revisions. This paper has come together well. A few final considerations are listed below.

On page 5 you specify western US yet on p 12 you say across the country. Please clarify and correct.

On page 13, COVID is discussed as was the relative trough in the geographic region at the time of this study. However, I believe this needs to be quantified regarding region of the country and local norms within the pandemic with regard to generalizability. This should be considered for inclusion in the limitations.

Top of page 14, "emphasis on" ?? Seems as though there is a missing word in this sentence

To clarify, were the 2 physicians who were excluded for low satisfaction scores the same two who were excluded due to burnout. Would consider specifying in the text.

In the response letter, you state that the participants were mostly mid-career. However, this assumes that none had come to medicine late or as a second career. Second, do you have a reference to support the statement that physicians of this age are a majority or "generally reflect the demographics of EM nationally"? There is some literature to support that mid-career in EM may be a particularly challenging time for women. Perhaps this should be included in the discussion.

Lall, Michelle D., et al. "Intention to leave emergency medicine: mid-career women are at increased risk." Western Journal of Emergency Medicine 21.5 (2020): 1131.

Lewiss, Resa E., et al. "Is academic medicine making mid-career women physicians invisible?." Journal of Women's Health 29.2 (2020): 187-192.

Reviewer #3: Thank you for the opportunity to review this manuscript. The authors interviewed 23 emergency physicians in the United States about wellness, developing a contextual framework explaining qualities associated with thriving.

Major points

The results section reads somewhat as a list of themes. The quotes are very short and there is little text. You interviewed 23 physicians so must have a number of examples for each theme. I think the results section should be expanded further to explain the theme topic, theme examples and why this theme explains how or why these physicians are thriving. If word limitations currently prevent this, much of the discussion could be reduced to allow for a fuller account of the study findings.

Minor points

Methods

How did you advertise for the participants? Did the advert ask for ‘resilient’ physicians? What wording was used in the invite? This is important as it will have impacted the type of person who responded to the advert.

I also wonder whether snowball sampling might have encouraged similar personalities to participate (in other words, friends who hung out together participated). This could have narrowed the range of personality traits and interests?

How many people responded to the advert? Were all people who responded screened and subsequently interviewed if they met inclusion criteria?

The manuscript states ‘Those who scored seven or higher for both questions were included in the study, as this correlates with satisfaction levels one standard deviation about the mean or greater’. There is a type error here (should read ‘above the mean’). In the results section you state ‘career satisfaction it was 7.46 [sd ± 2.21]’. If all participants had to score > 7 on the career satisfaction score then this summary result is not possible, since many would have score < 7.

Spirituality and religion is mentioned as a theme, however the quotes do not support this. I suggest either including supporting quotes or else removing the reference to both. The discussion also refers to spirituality and religion. This reference should be removed entirely unless the authors can substantiate the finding.

Since you interviewed only 23 physicians, you cannot state your participants reflect the diversity of the national workforce. If you are referring to ethnic or cultural diversity, you should state that.

7. PLOS authors have the option to publish the peer review history of their article (what does this mean?). If published, this will include your full peer review and any attached files.

Reviewer #1: No

Reviewer #3: **Yes: **Kerstin de Wit

---

## [Author Response · Author response to Decision Letter 2]

26 Aug 2022

RESPONSE TO REVIEWERS

Reviewer #1: Thank you for your additional revisions. This paper has come together well. A few final considerations are listed below.

On page 5 you specify western US yet on p 12 you say across the country. Please clarify and correct.

For clarity, we have deleted “across the country” on page 12. 

On page 13, COVID is discussed as was the relative trough in the geographic region at the time of this study. However, I believe this needs to be quantified regarding region of the country and local norms within the pandemic with regard to generalizability. This should be considered for inclusion in the limitations.

Will amend this part of the discussion to read, “These interviews were conducted during a relative trough in the COVID-19 pandemic at many of the hospitals. However, the pandemic still was prevalent during this study. We believe this makes these results even more compelling since these physicians met criteria for thriving and were not burned out during a pandemic.”

Top of page 14, "emphasis on" ?? Seems as though there is a missing word in this sentence

To keep this sentence parallel, the line now reads, “In particular, optimizing the work environment with appropriate resources and workload as well as emphasizing work-place culture through responsive leadership and social support were key drivers in our model.”

To clarify, were the 2 physicians who were excluded for low satisfaction scores the same two who were excluded due to burnout. Would consider specifying in the text.

This is correct. We highlight this in our results section, “Two physicians contacted by the interviewer met criteria for burnout and were not included in the study, leaving 23 for data analysis.”

In the response letter, you state that the participants were mostly mid-career. However, this assumes that none had come to medicine late or as a second career. Second, do you have a reference to support the statement that physicians of this age are a majority or "generally reflect the demographics of EM nationally"? There is some literature to support that mid-career in EM may be a particularly challenging time for women. Perhaps this should be included in the discussion.

Lall, Michelle D., et al. "Intention to leave emergency medicine: mid-career women are at increased risk." Western Journal of Emergency Medicine 21.5 (2020): 1131.

Lewiss, Resa E., et al. "Is academic medicine making mid-career women physicians invisible?." Journal of Women's Health 29.2 (2020): 187-192.

We include three references that suggest the demographics in this study generally reflect the demographics of national sample [10, 11, 12]. The concern about the diversity of our study sample came from PLOS ONE reviewers. Usually, qualitative studies are not designed to reflect larger demographics of a population, but are designed to elicit organic themes to build a model. In our case, this group generally reflected national gender and ethnic demographics. However, given that this was a qualitative study, and not a survey, we cannot make a statistical comparison. 

We have incorporated the Lall article into our discussion with the line, “Exploring a model of physician thriving is important, especially since many emergency physicians consider leaving the field, especially women in mid-career [24].”

Please note, this was not a study of burnout or those intending to leave practice. Rather, this was a study to build a conceptual model about what a thriving physician might look like. 

Reviewer #3: Thank you for the opportunity to review this manuscript. The authors interviewed 23 emergency physicians in the United States about wellness, developing a contextual framework explaining qualities associated with thriving.

Major points

The results section reads somewhat as a list of themes. The quotes are very short and there is little text. You interviewed 23 physicians so must have a number of examples for each theme. I think the results section should be expanded further to explain the theme topic, theme examples and why this theme explains how or why these physicians are thriving. If word limitations currently prevent this, much of the discussion could be reduced to allow for a fuller account of the study findings.

We have added six more quotations throughout the manuscript, including for the EMR, introverted-extroverted, administrative tasks and a balanced perspective. In general, most qualitative studies highlight 2-3 quotes per theme. There are multiple examples, but here are a few similar to ours from PLOS ONE:

Assessment of prognosis by physicians involved in work disability evaluation: a qualitative study. https://doi.org/10.1371/journal.pone.0212276

Surgeons’ emotional experience of their everyday practice – a qualitative study. https://doi.org/10.1371/journal.pone.0143763

What constitutes responsiveness of physicians: a qualitative study. https://doi.org/10.1371/journal.pone.0189962

What matters when doctors die: a qualitative study of family perspectives. https://doi.org/10.1371/journal.pone.0235138

Minor points

Methods

How did you advertise for the participants? Did the advert ask for ‘resilient’ physicians? What wording was used in the invite? This is important as it will have impacted the type of person who responded to the advert.

Since this was a study on physician well-being and thriving, the email and text invited physicians “to participate in a study on physician well-being and thriving.” We have added this phrase for clarification. We did not use the word “resilient” since this is a different concept, but may certainly influence well-being. In qualitative methodology, the snowball sampling technique is a valid, appropriate strategy for garnering participants. If a participant did not meet our criteria of life and job satisfaction, and was burned out, they were not included in our study. 

I also wonder whether snowball sampling might have encouraged similar personalities to participate (in other words, friends who hung out together participated). This could have narrowed the range of personality traits and interests?

As discussed above, snowball sampling is an appropriate recruitment strategy, especially for this type of study. Among the 23 participants, there were 10 hospitals represented, which suggests a broad sampling.

How many people responded to the advert? Were all people who responded screened and subsequently interviewed if they met inclusion criteria?

Through snowball and responding to the company’s email, 25 were screened, and 23 met criteria for the study. Since this was a qualitative study to describe a novel model of thriving, and not a survey, the denominator is not as important. 

The manuscript states ‘Those who scored seven or higher for both questions were included in the study, as this correlates with satisfaction levels one standard deviation about the mean or greater’. There is a type error here (should read ‘above the mean’). In the results section you state ‘career satisfaction it was 7.46 [sd ± 2.21]’. If all participants had to score > 7 on the career satisfaction score then this summary result is not possible, since many would have score < 7.

Indeed, you are correct. I double checked our statistics, the standard deviation was 1.21. Thanks for catching this. 

Spirituality and religion is mentioned as a theme, however the quotes do not support this. I suggest either including supporting quotes or else removing the reference to both. The discussion also refers to spirituality and religion. This reference should be removed entirely unless the authors can substantiate the finding.

In the discussion section, we explicitly state that religion and spirituality were not associated with our thriving model. In particular, we write, “Of note, our participants did not cite organized religious involvement and other formal social clubs as important to well-being, which has been noted in broader population-based studies [39].” Since religion and spirituality are often cited as supportive factors in other studies [we cite this in 40-42], we found this interesting and worthy of discussion. 

Since you interviewed only 23 physicians, you cannot state your participants reflect the diversity of the national workforce. If you are referring to ethnic or cultural diversity, you should state that.

This was in response to questions around diversity of our participants from the previous review. We have clarified the statement to read, “…which generally reflect the ethnic diversity of the national emergency physician workforce [17-19].”

---

## [Decision Letter · Decision Letter 3]

11 Oct 2022

PONE-D-22-01345R3Job and Life Satisfaction Among Emergency Physicians: a Qualitative StudyPLOS ONE

Dear Dr. Doolittle,

Thank you for submitting your manuscript to PLOS ONE. After careful consideration, we feel that it has merit but does not fully meet PLOS ONE’s publication criteria as it currently stands. Therefore, we invite you to submit a revised version of the manuscript that addresses the points raised during the review process.

ACADEMIC EDITOR: Thank you for addressing the reviewers' comments. There are some additional minor comments to address. Reviewer 1 has suggested providing more quotes to balance the distribution of quotes across the themes. I would also suggest providing more narrative description to explain the themes in richer detail. If you're struggling to find additional suitable quotes perhaps you could rely more on narrative description of the existing quotes. I know this is an open ended request so I will add some boundaries - perhaps aim for 3-5 additional descriptive sentences per theme. The reviewer has also suggested briefly discussing the limitations of snowball sampling. This is an important point to address.  An additional very important consideration is to anonymize the transcript that you report in your supplement. There are several instances in which the position and institution of your participant is reported, which would allow identification of the person. Please eliminate or redact any identifying information to fall in compliance with your ethical approval, which I assume does not allow identifying information to be reported.  Be sure to:Indicate which changes you require for acceptance versus which changes you recommendAddress any conflicts between the reviews so that it's clear which advice the authors should followProvide specific feedback from your evaluation of the manuscriptPlease ensure that your decision is justified on PLOS ONE’s publication criteria and not, for example, on novelty or perceived impact.

We look forward to receiving your revised manuscript.

Kind regards,

Niklas Bobrovitz

Academic Editor

PLOS ONE

Journal Requirements:

Reviewers' comments:

Reviewer's Responses to Questions

**Comments to the Author**

1. If the authors have adequately addressed your comments raised in a previous round of review and you feel that this manuscript is now acceptable for publication, you may indicate that here to bypass the “Comments to the Author” section, enter your conflict of interest statement in the “Confidential to Editor” section, and submit your "Accept" recommendation.

Reviewer #1: (No Response)

2. Is the manuscript technically sound, and do the data support the conclusions?

Reviewer #1: Partly

3. Has the statistical analysis been performed appropriately and rigorously? 

Reviewer #1: N/A

4. Have the authors made all data underlying the findings in their manuscript fully available?

Reviewer #1: Yes

5. Is the manuscript presented in an intelligible fashion and written in standard English?

Reviewer #1: Yes

6. Review Comments to the Author

Reviewer #1: Thank you for your revisions.

The results section still needs work. Some areas have multiple quotes and others do not. Considering the qualitative nature of the paper, additional quotes are needed to support the themes. The richness of the quotes will amplify the themes and make them more interesting.

A comment about snowball sampling, it is appropriate for the methodology but does have limitations which should be mentioned. If like finds like and more engaged and extroverted people are responding and passing along to similar people, there is potential for bias in sampling. This should be commented on.

7. PLOS authors have the option to publish the peer review history of their article (what does this mean?). If published, this will include your full peer review and any attached files.

Reviewer #1: No

---

## [Author Response · Author response to Decision Letter 3]

18 Nov 2022

Author Response To Reviewers

Reviewer #1: Thank you for your revisions.

The results section still needs work. Some areas have multiple quotes and others do not. Considering the qualitative nature of the paper, additional quotes are needed to support the themes. The richness of the quotes will amplify the themes and make them more interesting.

Many thanks for your continued support of our manuscripts. 

We have included several additional quotes throughout to support our findings. Each domain has 4-6 quotations. This feels about right and is consistent with similar qualitative studies. 

A comment about snowball sampling, it is appropriate for the methodology but does have limitations which should be mentioned. If like finds like and more engaged and extroverted people are responding and passing along to similar people, there is potential for bias in sampling. This should be commented on.

While snowball sampling has its limits, this was the proper strategy for this project, given the specialized subpopulation (ie. not burned out AND also satisfied with life and their job) (doi: 10.1111/j.1442-2018.2010.00541.x), we do acknowledge the limitations, and include the following line, “While snowball sampling was the appropriate technique for reaching this specialized subpopulation, limitations should be acknowledged. Namely, these participants were a non-random sample. Physicians may have referred participants in their social network, with similar beliefs and ideas. Thus, there may have been themes that were not captured. However, given the diversity of the participants, with ten hospitals represented, and thematic saturation after ten interviews, we believe our sample population represented valid findings.”

---

## [Editor Report · Decision Letter 4]

7 Dec 2022

Job and Life Satisfaction Among Emergency Physicians: a Qualitative Study

PONE-D-22-01345R4

Dear Dr. Doolittle,

We’re pleased to inform you that your manuscript has been judged scientifically suitable for publication and will be formally accepted for publication once it meets all outstanding technical requirements.

Kind regards,

Niklas Bobrovitz

Academic Editor

PLOS ONE
---

## [Editor Report · Acceptance letter]

19 Dec 2022

PONE-D-22-01345R4 

Job and Life Satisfaction Among Emergency Physicians: a Qualitative Study 

Dear Dr. Doolittle:

I'm pleased to inform you that your manuscript has been deemed suitable for publication in PLOS ONE. Congratulations! Your manuscript is now with our production department. 

Kind regards, 

on behalf of

Dr. Niklas Bobrovitz 

Academic Editor

PLOS ONE